# Prevalence and Course of SARS-CoV-2 Infection among Immunocompromised Children Hospitalised in the Tertiary Referral Hospital in Poland

**DOI:** 10.3390/jcm10194556

**Published:** 2021-09-30

**Authors:** Karolina Kuczborska, Janusz Książyk

**Affiliations:** Department of Pediatrics, Nutrition and Metabolic Disorders, Children’s Memorial Health Institute, 04-730 Warsaw, Poland; j.ksiazyk@ipczd.pl

**Keywords:** cancer, liver transplant, immunodeficiency, paediatric population, SARS-CoV-2

## Abstract

The aim of this retrospective study was the assessment of the prevalence, clinical characteristics, and the comparison of the course of SARS-CoV-2 infection in children with and without immunodeficiency that were admitted to the COVID-19 Subunit of the tertiary referral hospital in Warsaw, Poland. We analysed 313 children hospitalised in the COVID-19 Subunit. The analysis was performed on the total study group and subgroups of children with and without immunodeficiency. In each group, clinical data and laboratory test results were analysed. Of the 68 children with isolated fevers, only seven (11.76%) were diagnosed with COVID-19, whereas among those with the accompanying features of respiratory or gastrointestinal infection, only one (3.23%) and ten (16.95%) patients tested positive, respectively. In both groups, the course of the infection was mainly asymptomatic or mild. The children with immunodeficiencies had lower white blood cell and lymphocyte counts, lower haemoglobin levels, and higher urea levels but did not differ in other biochemical variables. To conclude, the most frequently reported symptoms of COVID-19 indicate that this disease among children is only a small percentage. In both groups, the responses to the infection were comparable in terms of the mild clinical symptoms and the laboratory test results. Therefore, SARS-CoV-2 infection should not alter the chronic treatment of underlying diseases.

## 1. Introduction

The first reports of COVID-19 appeared at the end of December 2019. The first cases in Europe were described on 24 January 2020 [1]. To date, the pandemic has spread worldwide, affecting almost 230 million people and causing more than 4.5 million deaths [2].

Since the beginning of the pandemic, a mild or asymptomatic course of infection was noticeable in children [3]. Most contracted it from their parents or other adult close contacts [4]. Unlike adults, however, children much less frequently required hospitalisation or specialist treatments, a phenomenon that various researchers have tried to explain. These researchers have proved that a severe course of interstitial pneumonia in adults was caused by a tissue-damaging excessive inflammatory response and a cytokine storm syndrome. However, children were assumed to have a different inflammatory response, with a higher number of regulatory T and B lymphocytes involved in immune tolerance, which reduces the body’s inflammatory response to infections [5]. Children were also characterised by a higher basal expression of relevant pattern recognition receptors in the upper airway epithelial cells, the dendritic cells, and the macrophages, which guaranteed a stronger innate response to viral infections [6]. Other causes contributing to the milder cases of COVID-19 in children may be a lower expression of angiotensin-converting enzyme 2 (ACE2) receptors by which the virus enters cells, fewer comorbidities, and the fact that children are more likely to be nonsmokers [7,8].

Nevertheless, despite the mild course of the infection in the general paediatric population, it has been postulated that immunosuppression may harm the clinical picture of the disease.

Therefore, this study evaluated the prevalence, clinical characteristics, and the comparison of the course of SARS-CoV-2 infection in children with and without immunodeficiency that were admitted to the COVID-19 Subunit of the tertiary referral hospital in Warsaw, Poland, during the second and third waves of the COVID-19 pandemic.

## 2. Materials and Methods

### 2.1. Patients

This retrospective study analysed 313 children (136 girls and 176 boys) hospitalised in the COVID-19 Subunit in the Department of Paediatrics, Nutrition, and Metabolic Disorders, Children’s Memorial Health Institute in Warsaw, Poland, between 1 November 2020 and 30 April 2021, which corresponds to the periods of the second and the third waves of the COVID-19 pandemic.

The study protocol was approved by the institutional Ethics Committee at the Children’s Memorial Health Institute in Warsaw, Poland.

### 2.2. Admission Criteria

The admission criteria to the COVID-19 Subunit in the Department of Paediatrics, Nutrition, and Metabolic Disorders were as follows:Symptomatic COVID-19 in children under constant medical care at Children’s Memorial Health Institute requiring hospitalisation;Asymptomatic COVID-19 or quarantine in children under constant medical care at Children’s Memorial Health Institute requiring specialised diagnostics or treatment which could not be postponed prior to recovery from the virus;Symptoms of suspected COVID-19 in all children admitted urgently to the Children’s Memorial Health Institute. (The hospitalisation in the COVID-19 Subunit aimed to exclude SARS-CoV-2 infection before the possible transfer to the appropriate ward.)

### 2.3. Diagnosis of SARS-CoV2 Infection

All the children and accompanying parents had a nasopharyngeal swab collected at the beginning of the hospitalisation in order to perform the RT-PCR test for SARS-CoV-2.

Persons excluded from swab collection were:Those diagnosed with the infection before hospitalisation;Convalescents with a history of illness within the last three months.

### 2.4. Division of the Subgroups

We analysed the total study group and subgroups of children with (*n* = 110, 35.14%) and without (*n* = 203, 64.86%) immunodeficiency. Among the immunodeficiency factors, we included: tumours, liver and kidney transplantation, demyelinating diseases, ulcerative colitis, end-stage renal disease, asplenia, Wiskott–Aldrich syndrome, and DiGeorge syndrome. Some children had more than one immunodeficiency factor.

### 2.5. Clinical and Laboratory Data

In all patients, we analysed: Clinical data: age, sex, the reasons for admission, COVID-19 status in the patient and the caregiver, final diagnosis, the manifestation of the disease, treatment, incidence of complications, comorbidities, and the presence of immunodeficiency;Body temperature, blood oxygen saturation, respiratory rate, heart rate, and blood pressure on hospital admission;Laboratory results on admission: white blood cell (WBC) count, neutrophil-to-lymphocyte ratio (NLR), C-reactive protein (CRP), procalcitonin (PCT), erythrocyte sedimentation rate (ESR), level of haemoglobin (Hgb) and platelets (PLT), D-dimers, fibrinogen, lactate dehydrogenase (LDH), transaminases (ALT and AST), creatinine and urea levels, and N-terminal prohormone of brain natriuretic peptide (NT-proBNP);Chest X-ray results.

### 2.6. Fever—Definitions 


Fever—A body temperature above 38 °C;Isolated fever—A body temperature above 38 °C without any other accompanying symptoms;Febrile neutropenia—A body temperature greater than or equal to 38 °C for at least an hour, with an absolute neutrophil count of less than 1.5 × 10^9^ cells/L.


### 2.7. Disease Severity—Definitions


Asymptomatic—No symptoms of COVID-19;Mild—Symptoms that generally did not require hospitalisation (i.e., moderate fever or cough that lasted for several days, rhinorrhoea, moderate gastrointestinal symptoms, rash, etc.). However, hospitalisation was required for other indications;Moderate—Required hospitalisation in a paediatric ward;Severe—Required hospitalisation in the ICU;PIMS (Paediatric Inflammatory Multisystem Syndrome)—The diagnosis was based on WHO diagnostic criteria [9].


### 2.8. Statistical Analysis

The nonparametric Mann–Whitney U-test was used to compare the quantitative variables, whose distribution was significantly different from the Gaussian distribution. The chi-square test was used for comparing the qualitative data. The probability value of *p* < 0.05 was considered statistically significant. Microsoft Excel and Statistica 13 were used for the analysis.

## 3. Results

### 3.1. Entire Study Group

Between November 2020 and April 2021, 313 children were hospitalised in the COVID-19 Subunit in the Department of Paediatrics, Nutrition, and Metabolic Disorders, Children’s Memorial Health Institute in Warsaw, Poland. The age of the patients ranged from 17 days to 17 years of age (median 3.00; Q1 = 1.00, Q3 = 9.00). Table 1 shows the reasons for admission.

From the entire study group, 105 (33.55%) children had to remain in the Subunit due to confirmed COVID-19 results or because they had been placed in quarantine (Figure 1). 

### 3.2. Children with Immunodeficiency

There were 110 (35.14%) children with immunodeficiency hospitalised in the COVID-19 Subunit. Tumours (due to chemotherapy) were the most common contributing immunodeficiency factor in this group (*n* = 78, 70.91%), the majority of which were central nervous system tumours (*n* = 21, 26.92%), neuroblastomas (*n* = 11; 14.10%), medulloblastomas and retinoblastomas (*n* = 8, 11.11% each). The other most common immunosuppressive states were the result of liver or kidney transplantations (*n* = 23, 23.64% and *n* = 3, 2.72% respectively).

The main causes of patients’ admission to the hospital were: scheduled chemotherapy (*n* = 39, 35.45%), fever (*n* = 28, 25.45%), chemotherapy side effects (*n* = 6, 5.45%) and features of respiratory tract infection (*n* = 5, 4.55%). 

### 3.3. Comparison of Groups of Children with and without Immunodeficiency

#### 3.3.1. Symptoms on Admission

Children with and without immunodeficiency were often admitted to the hospital for slightly different reasons; however, isolated fever prevailed in both groups (*n* = 28, 25.45% vs. *n* = 40, 19.7%). 

Of the 68 children who had isolated fevers, only eight (11.76%) were diagnosed with COVID-19, and the difference in this percentage in both groups was not statistically significant. Among the children with immunodeficiency, febrile neutropenia was the principal diagnosis, whereas urinary tract infections were more common among those with a normal immune system (Table 2).

Among the 31 children who had the accompanying features of respiratory tract infection (i.e., cough, dyspnoea, and rhinorrhoea), only one (3.23%) was diagnosed with SARS-CoV-2 (Table 3). In children with and without immunodeficiency, pneumonia and upper respiratory tract infections of a different aetiology were the most prevalent to a comparable degree.

Of the 59 patients presenting symptoms of gastrointestinal infection, only ten were diagnosed with COVID-19, which was a similar percentage in both groups. Nevertheless, most (*n* = 6) were diagnosed with other aetiological factors, such as rotavirus, norovirus, and adenovirus, or Clostridium difficile (Table 4).

#### 3.3.2. COVID-19 Course

Among those children with a confirmed COVID-19 diagnosis, the most common symptom on admission was a fever. However, this occurred in only 25% of the cases, which was comparable in both groups. Of the commonly reported symptoms, only diarrhoea and vomiting predominated significantly in the immunocompetent children (Table 5). 

The course of the infection in the subgroups was similar. Both children with and without immunodeficiency were mainly asymptomatic or had a mild course of COVID-19. Only one immunocompromised child with sarcoma experienced a severe disease course that required oxygen therapy and hospitalisation in the ICU Department. Moreover, in two immunocompetent children, the severe course of the disease was more due to an underlying condition (newly diagnosed diabetes) than to COVID-19. PIMS affected children without immunodeficiency significantly more often (Table 5).

In laboratory test results on admission, children with immunodeficiency had lower white blood cell counts, lower lymphocyte counts, and, therefore, a higher neutrophil-to-lymphocyte ratio (NLR). They also presented lower haemoglobin levels and higher urea levels. However, children diagnosed with SARS-CoV-2 infection with and without immunodeficiency did not differ in terms of the level of inflammatory markers (i.e., CRP, PCT, and ESR), nor in terms of other tested biochemical variables (Table 6).

Chest X-rays performed on children with symptomatic COVID-19 (i.e., fever, cough, and dyspnoea) revealed inflammatory changes in only five of them (19.23%), with one immunocompromised and four immunocompetent.

Most of the oncology patients hospitalised in the COVID-19 Subunit with SARS-CoV-2 infections were admitted so to undergo chemotherapy. We neither had to delay therapies nor did we notice a worse tolerance of the treatment. We also performed scheduled radiotherapy in two children without any additional complications. Moreover, even during symptomatic COVID-19, we did not modify the immunosuppressive treatment in transplant patients if coexisting acute states did not require it. Moreover, the monoclonal antibodies were administered to patients with atypical haemolytic uremic syndrome (Eculizumab; one patient) and ulcerative colitis (Adalimumab; one patient). Patients with demyelinating diseases received immunoglobulins with glucocorticoids (Devic’s Disease; one patient) or glucocorticoids (multiple sclerosis; one patient) as scheduled. A good tolerance of the treatment was noticeable in each of the above-mentioned cases.

## 4. Discussion

Although the pandemic has lasted for a year and a half, reports of cases concerning immunocompromised children are still relatively scarce. This population may be less frequently exposed to SARS-CoV-2 due to the increased isolation that is crucial to their therapies, and as a result, may not become a part of regular studies [10]. Even in the Children’s Memorial Health Institute in Warsaw, Poland’s biggest tertiary referral hospital, we were only able to collect data on 55 immunocompromised, SARS-CoV-2-infected patients during the six-month observation period.

The most frequently reported symptoms of COVID-19 in children included: fever, cough, sore throat, gastrointestinal symptoms (i.e., vomiting/diarrhoea), myalgia, and fatigue [5]. However, these symptoms are non-specific and may indicate other infectious or non-infectious diseases. Fever without an identifiable cause (<7 days’ duration) among immunocompetent children was mainly caused by upper and lower respiratory tract or urinary tract infections [11], whereas among immunocompromised children, it could also be brought on by febrile neutropenia. We have proved that the pandemic did not significantly alter this trend and that COVID-19 can be one but not the most common cause of fever in children. The issue of acute coughs looks similar, which are primarily caused by respiratory tract infections [12] of viral aetiology rather than SARS-CoV-2 [13]. Having studied the hospitalised population, there is a higher incidence of pneumonia than of upper respiratory tract infections. Nevertheless, in both studied groups with respiratory symptoms, COVID-19 was one of the rarest diagnoses. SARS-CoV-2 occasionally accompanies gastrointestinal symptoms, but even if so, usually other and more specific viruses are detected. 

The dominant population in the study group with immunodeficiency were children with tumours. Almost all of them had an asymptomatic course of the disease or had mild symptoms. Due to this fact, we did not delay their therapies and proceeded with chemotherapy or radiotherapy. In their study from the UK, Millen et al. also reported the prevalence of a mild or asymptomatic course of COVID-19 in their cancer patients (91%), and they also did not feel the need for any significant delays in treatments [14]. In the study from Lombardia conducted at the beginning of the pandemic, the researchers reported only two cases of pneumonia (one mild and one more severe) out of 42 positive patients in the six paediatric onco-haematology centres [15]. Among the available studies, only researchers from New York presented less optimistic data. In their population, 74.5% of oncological patients had a symptomatic COVID-19 course, 25.5% required oxygen therapy, and 7% required mechanical ventilation. They also proved that there is a higher risk of severe disease in obese males. Moreover, a predominance of Hispanic/Latino ethnicities was observed in the group [16], leading to the suspicion of an increased risk of disease severity in this group [17]. Perhaps this is the cause of the entirely different results than from studies performed elsewhere. This hypothesis requires further study. 

The second dominant group among our patients were liver and kidney transplant recipients. Even those who had coexisting PTLD did not experience a severe course of COVID-19. They were admitted either for chemotherapy or due to complications from the transplantation, such as gastrointestinal bleeding or cholangitis. No immunosuppression scheme was altered during SARS-CoV-2 infection if coexisting acute states did not require it. Furthermore, during the asymptomatic course of the disease, we also administered eculizumab in a patient after kidney transplantation with good tolerance. Our results are consistent with those obtained in a large study from Bergamo. They proved that paediatric liver transplant recipients either developed mild symptoms or did not present any symptoms despite having household contact with infected relatives. None of the patients required hospitalisation or suffered from pneumonia [18]. Based on scarce data, we assumed that immunosuppression, as an individual factor, may not worsen the course of COVID-19. On the contrary, data from the European Liver Transplant Registry suggest a protective effect of tacrolimus on disease development in adults [19]. 

All three children hospitalised in our department with ulcerative colitis remained asymptomatic throughout the course of infection, and there was no delay in their treatment. Unfortunately, the literature lacks data on the course of SARS-CoV-2 infection in this study group. Therefore, the researchers focused mainly on patients’ treatment adherence or changes to in-hospital care [20,21]. However, a cohort study from Sweden reports that among adults with Inflammatory Bowel Disease (IBD), despite the risk of hospital admission during infection being higher, the course of the disease is not more severe than in the general population [22].

There may be several reasons for the mild course of infection in children with immunodeficiency. Some authors have reported that increased isolation and sanitation may be a possible explanation. However, these measures reduce the risk of infection rather than the severity of its course. A more important factor seems to be immunosuppression itself, which limits the inflammatory process responsible for a severe COVID-19 pattern [23]. Therefore, the factor that usually predisposes to more frequent and heavier infections is, paradoxically, a protective factor for COVID-19 [23]. The mechanisms of this phenomenon are better studied on patients with primary rather than secondary immunodeficiency diseases. Researchers in this area reported that children with agammaglobulinemia tend to have a milder course of the disease, suggesting that B-cell immunity does not play a key role in the defence against the virus, nor is it responsible for the SARS-CoV-2-induced hyper inflammation process [24,25,26]. Immunosuppression mechanisms in children with secondary immunodeficiency are more complex; however, they usually impair the B-cell’s function. Nevertheless, further studies are required in this field. 

The role of numerous biomarkers in the diagnostic, prognostic, and management process of SARS-CoV-2 infection has been carefully studied by researchers. A hallmark of the disease is leukopenia with lymphocytopenia, which correlates with disease severity. Regarding inflammatory markers, the CRP level is significantly more frequently elevated than PCT. Other laboratory markers that often show elevated levels among children include D-dimer, creatine kinase, transaminases, and urea. Decreased haemoglobin levels are observed mainly in a severe COVID-19 pattern [27]. In our patients, neither leucocytosis nor leukopenia was common. While CRP was a more sensitive marker of inflammation than PCT, it did not reach high levels. Thus, we did not observe that any biochemical markers were a hallmark of the infection. Concerning the differences between the groups, a lower count of lymphocytes, a lower haemoglobin level, and a higher level of urea among immunocompromised children, in our view, may be due to the underlying diseases of this group rather than to a different response to the infection.

## 5. Conclusions

The most frequently reported symptoms of SARS-CoV-2 infection, such as fever, cough, dyspnoea, or gastrointestinal symptoms, even during the second and third waves of COVID-19, indicate that this disease among children is only a small percentage. Therefore, in both immunocompromised and immunocompetent children manifesting the above symptoms, testing to rule out SARS-CoV-2 infection should be always carried out; however, other diagnostics should not be delayed since there is a high probability that the cause is entirely different.

The course of the SARS-CoV-2 infection among children with immunodeficiency is mainly asymptomatic or mild and does not differ significantly from those who are immunocompetent. They may even have a reduced risk of PIMS due to the decreased activity of the immune system, which is crucial in this disease.

Immunosuppressive treatment or chemotherapy should not be withdrawn or delayed because of SARS-CoV-2 infection if coexisting acute statesdo not require it.

Immunocompromised children may present a lower count of leukocytes and lymphocytes and a lower haemoglobin level during COVID-19 compared to children with normal immune systems. However, these differences may be falsified by underlying diseases with coexisting acute conditions. In addition, no other tested laboratory marker seems to help differentiate the response to infection between the groups.

## Figures and Tables

**Figure 1 jcm-10-04556-f001:**
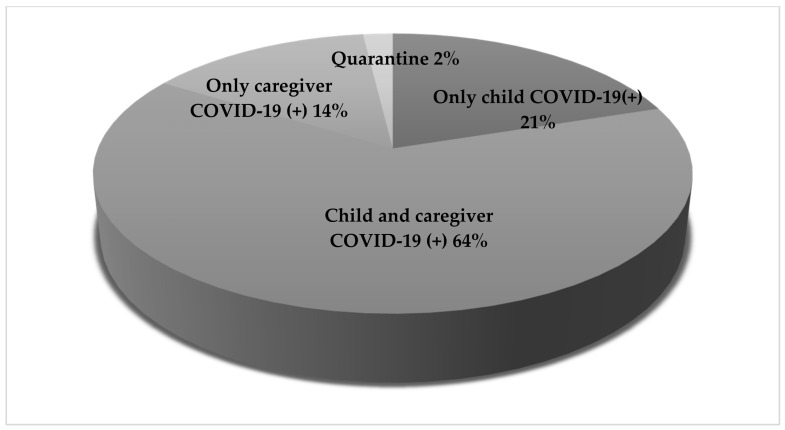
The COVID-19 status in children who remained in the COVID-19 Subunit.

**Table 1 jcm-10-04556-t001:** Reasons for hospital admission.

Reasons for Hospital Admission		
Without the initial diagnosis of SARS-CoV-2 infection		
Reason	Number	%
Isolated fever	68	21.73
Features of gastroenteritis	59	18.85
Features of respiratory infection (cough, dyspnoea)	31	9.90
Suspected urinary tract infection (UTI)	20	6.39
Suspected sepsis	18	5.75
Suspected Kawasaki Disease/PIMS	8	2.56
Poisoning	6	1.92
Acute abdomen	6	1.92
Epileptic seizures	4	1.28
Sinusitis	4	1.28
Stomatitis	2	0.64
Severe weakness	2	0.64
Declines in blood oxygen saturation	1	0.32
Lymphadenopathy	1	0.32
Other	15	4.79
With the initial diagnosis of SARS-CoV-2 infection		
Reason	Number	%
Oncological treatment (mainly CTH)	41	13.10
Newly diagnosed type 1 Diabetes (DM1)	8	2.56
Chemotherapy side effects	6	1.92
Post-transplant problems	6	1.92
Heart defects treatment	4	1.28
Gastrointestinal bleeding	4	1.28
Demyelinating diseases treatment	4	1.28
COVID-19	4	1.28
Neurosurgical problems	3	0.96
Ulcerative colitis treatment	2	0.64
Treatment or diagnosis of other diseases	13	4.15

Abbreviations: COVID-19—Coronavirus Disease; CTH—chemotherapy; DM1—type 1 diabetes mellitus; PIMS—Paediatric Inflammatory Multisystem Syndrome; UTI—urinary tract infection.

**Table 2 jcm-10-04556-t002:** Final diagnosis in children who had isolated fever.

Diagnosis	Entire Study Group *n* = 68		ID (+) *n* = 28		ID (−) *n* = 40		*p*
	*n*	%	*n*	%	*N*	%	
Febrile neutropenia	14	20.59	14	50.00	0	0.00	<0.01*
UTI	10	14.70	1	3.57	9	22.50	0.03*
FUO	8	11.76	3	10.71	5	12.50	0.20
COVID-19	8	11.76	2	7.14	6	15.00	0.47
URTI	6	8.82	3	10.71	3	7.50	0.65
Gastroenteritis	3	4.41	1	3.57	2	5.00	0.78
Sepsis	3	4.41	1	3.57	2	5.00	0.50
Pneumonia	3	4.41	0	0.00	3	7.50	0.14
PFAPA	2	2.94	0	0.00	2	5.00	0.23
Roseola	2	2.94	0	0.00	2	5.00	0.23
Cholangitis	2	2.94	1	3.57	1	3.50	0.80
PIMS	1	1.47	1	3.57	0	0.00	0.23
Mononucleosis	1	1.47	0	0.00	1	2.50	0.40
Sinusitis	1	1.47	0	0.00	1	2.50	0.40
Stomatitis	1	1.47	0	0.00	1	2.50	0.40
Acute pancreatitis	1	1.47	0	0.00	1	2.50	0.40
Postoperative wound infection	1	1.47	0	0.00	1	2.50	0.40
Scarlet fever	1	1.47	0	0.00	1	2.50	0.40
Parotitis	1	1.47	0	0.00	1	2.50	0.40

Abbreviations: COVID-19—Coronavirus Disease; FUO—fever of unknown origin; ID (+)—children with immunodeficiency; ID (−)—children without immunodeficiency; PFAPA—periodic fevers with aphthous stomatitis, pharyngitis, and adenitis; PIMS—Paediatric Inflammatory Multisystem Syndrome; URTI—upper respiratory tract infection; UTI—urinary tract infection. *—*p* values < 0.05.

**Table 3 jcm-10-04556-t003:** Final diagnosis in children who had respiratory symptoms.

Diagnosis	Entire Study Group *n* = 31		ID (+) *n* = 5		ID (−) *n* = 26		*p*
	*n*	%	*n*	%	*n*	%	
Pneumonia	17	54.84	2	40.00	15	57.69	0.41
URTI	10	32.26	2	40.00	8	30.77	0.69
Bronchiolitis	2	6.45	0	0.00	2	7.69	0.52
COVID-19	1	3.23	1	20.00	0	0.00	0.02*
Mononucleosis	1	3.23	0	0.00	1	3.85	0.65

Abbreviations: COVID-19—Coronavirus Disease; ID (+)—children with immunodeficiency; ID (−)—children without immunodeficiency; URTI—upper respiratory tract infection. *—*p* values < 0.05.

**Table 4 jcm-10-04556-t004:** Final diagnosis in children who had gastrointestinal symptoms.

Diagnosis	Entire Study Group *n* = 59		ID (+) *n* = 14		ID (−) *n* = 45		*p*
	*n*	%	*n*	%	*n*	%	
Gastroenteritis	34	57.63	5	35.71	29	64.44	0.63
COVID-19	10	16.95	4	28.57	6	13.33	0.17
UTI	8	13.56	5	35.71	3	6.67	0.26
Pneumonia	4	6.78	0	0.00	4	8.89	0.32
URTI	3	5.08	0	0.00	3	6.67	0.48
PIMS/Kawasaki disease	2	3.39	0	0.00	2	4.44	0.77
DM1	2	3.39	0	0.00	2	4.44	0.21
Sepsis	2	3.39	0	0.00	2	4.44	0.48
Acute abdomen	2	3.39	0	0.00	2	4.44	0.48
Gastrointestinal bleeding	2	3.39	0	0.00	1	2.22	0.27

Abbreviations: COVID-19—Coronavirus Disease; DM1—type 1 diabetes mellitus; ID (+)—children with immunodeficiency; ID (−)—children without immunodeficiency; PIMS—Paediatric Inflammatory Multisystem Syndrome; URTI—upper respiratory tract infection; UTI—urinary tract infection.

**Table 5 jcm-10-04556-t005:** Symptoms on admission and the course of COVID-19 in children with and without immunodeficiency.

	Entire Study Group, *n*= 88		ID (+) *n* = 55		ID (−) *n* = 33		*p*
Symptoms on Admission	*n*	%	*n*	%	*n*	%	
Fever	22	25.00	11	20	11	33.33	0.17
Cough	9	10.23	4	7.27	5	15.15	0.24
Dyspnoea	4	4.55	3	5.45	1	3.03	0.60
Rhinorrhoea	9	10.23	4	7.27	5	15.15	0.25
Diarrhoea	8	9.09	2	3.63	6	18.18	0.02*
Vomiting	10	11.36	3	5.45	7	21.21	0.02*
Weakness	3	3.41	2	3.63	1	3.03	0.13
Declines in blood oxygen saturation	2	2.27	2	3.63	0	0	0.27
Rash	2	2.27	1	1.81	1	3.03	0.29
Course of the disease	*n*	%	*n*	%	*n*	%	
Asymptomatic	58	65.91	40	72.72	19	57.58	0.55
Mild	17	19.32	12	21.81	5	15.15	0.44
Moderate	4	4.55	1	1.82	3	9.09	0.12
Severe	3	3.41	1	1.82	2	6.06	0.71
PIMS	5	5.68	1	1.82	4	12.12	0.04*

Abbreviations: ID (+)—children with immunodeficiency; ID (−)—children without immunodeficiency; PIMS—Paediatric Inflammatory Multisystem Syndrome. *—*p* values < 0.05.

**Table 6 jcm-10-04556-t006:** Comparison of the laboratory test results in children with and without immunodeficiency.

Variable		Q1	Median	Q3	Association Type	Association	*p*
WBC							
(×10^9^/L)	ID (+)	3.72	5.40	8.49	Negative	−2.65	0.01 *
	ID (−)	5.40	7.72	11.38			
NEUT							
(×10^9^/L)	ID (+)	1.49	2.32	3.96	N/A	−0.36	0.72
	ID (−)	1.52	2.30	4.14			
LYMPH							
(×10^9^/L)	ID (+)	0.97	1.40	3.23	Negative	−3.75	<0.01 *
	ID (−)	2.06	4.00	5.78			
NLR							
	ID (+)	0.49	1.16	3.44	Positive	2.07	0.04 *
	ID (−)	0.37	0.58	1.17			
HGB							
(g/dL)	ID (+)	9.90	10.80	12.55	Negative	−1.98	0.05 *
	ID (−)	10.75	12.00	13.10			
PLT							
(×10^9^/L)	ID (+)	182.00	281.00	395.50	N/A	1.32	0.19
	ID (−)	231.50	319.00	433.5			
CRP							
(mg/dL)	ID (+)	0.07	0.10	2.18	N/A	0.41	0.68
	ID (−)	0.09	0.10	0.80			
PCT							
(ng/mL)	ID (+)	0.04	0.08	0.50	N/A	−0.39	0.69
	ID (−)	0.03	0.13	0.79			
ESR							
(mm/h)	ID (+)	16.00	21.00	55.00	N/A	−0.18	0.85
	ID (−)	35.00	63.00	64.50			
D-dimers							
(µg/L)	ID (+)	194.00	426.50	2035.50	N/A	0.69	0.49
	ID (−)	566.00	950.00	1264.00			
Fibrinogen							
(g/L)	ID (+)	2.44	2.85	3.86	N/A	−1.06	0.29
	ID (−)	2.07	2.30	3.42			
LDH							
(U/L)	ID (+)	248.00	315.50	386.75	N/A	−0.49	0.63
	ID (−)	225.50	288.50	306.00			
ALT							
(U/L)	ID (+)	13.75	18.00	61.50	N/A	0.02	0.98
	ID (−)	13.25	25.00	31.50			
AST							
(U/L)	ID (+)	23.00	27.50	35.25	N/A	0.87	0.38
	ID (−)	22.50	34.00	40.00			
Creatinine							
(mg/dL)	ID (+)	0.38	0.47	0.61	N/A	−0.77	0.44
	ID (−)	0.40	0.44	0.53			
Urea							
(mg/dL)	ID (+)	16.18	22.15	30.75	Positive	2.05	0.04 *
	ID (−)	14.45	19.70	23.50			
NT-proBNP							
(pg/mL)	ID (+)	347.15	399.40	403.85	N/A	−1.31	0.19
	ID (−)	118.83	138.90	232.15			

Abbreviations: ALT—alanine aminotransferase; AST—aspartate aminotransferase; CRP—C-reactive protein; ESR—erythrocyte sedimentation rate; HGB—haemoglobin level; ID (+)—children with immunodeficiency; ID (−)—children without immunodeficiency; LDH—lactate dehydrogenase; NLR—neutrophil-to-lymphocyte ratio; NT-proBNP—N-terminal pro-B-type natriuretic peptide; PCT—procalcitonin; PLT—platelet count; WBC—white blood cell count. *—*p* values < 0.05.

## Data Availability

The data presented in this study are available on request from the corresponding author.

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
