# Peer review of "Prevalence and Course of SARS-CoV-2 Infection among Immunocompromised Children Hospitalised in the Tertiary Referral Hospital in Poland"

_jcm, 2021, doi:10.3390/jcm10194556_

Round 1
Reviewer 1 Report
Dear authors, I was honored to review your manuscript entitled "Prevalence and course of SARS-CoV-2 infection among im-munocompromised children hospitalised in the tertiary referral hospital in Poland".
This is such an interesting manuscript that will bring light to the fact that IS children did not present with higher morbidity or mortality with regards to COVID-19 when compared to the general population.
However, there is a major point that, in my opinion, needs to be readressed before reviewing again your manuscript: only patients with confirmed SARS-CoV-2 infection (acute COVID-19 or MIS-C) should be included in the analysis. Otherwise, your -otherwise appropriate- local approach to the management of suspected cases causes a severe bias. Once this is done, all the sections should be appropriately rewritten.
Moreover, there are some other, mostly minor, comments that may be considered before publication, if finally accepted. See them listed below:
Introduction is well written. Please update the number of total infections and deaths worldwide. I'd change parents or other relatives to parents or other adult close contacts. Please add the citation https://www.nature.com/articles/s41587-021-01037-9 regarding different immune responses in children. I'd be more careful when describing other reasons for a milder course of the disease: change are to may be or similar.
From methods section to discussion: please see my major comment. In my opinion, once this is changed, deep review may be performed. In fact, section 3.3.2 is key in the results section and should be extended together with lab results only in those with confirmed SARS-CoV-2 infection.
Reviewer 2 Report
This study reports interesting data concerning children screened for COVID-19 in Poland.
Some additional clarification and correction should be made:
- because your research involved human subjects, please add the ethical approval committee and approval code to the section of Materials and Methods.
- please rephrase, it is unclear: ʺTo conclude, in both groups of children the most frequent symptoms of COVID-19 the diagnosis is highly probable entirely different. Moreover, the response to the infection is comparable in terms of mild clinical symptoms and laboratory test resultsʺ.
- please mention in Materials and Methods section what is considered fever, isolated fever (on admission), febrile neutropenia in this study.
- there are differences between the number of patients with FUO in table 2 (8 patients) and the number of patients mentioned in the description of this table (seven).
- formatting of the citations is not consistent throughout the manuscript (see introduction and discussion section).
- Please include the units of measure for the laboratory parameters in Table 6.
Other observations could be also interesting:
- It would be useful to have a table describing patients with immunodeficiency according to the associated factors: (i) for the whole study group, (ii) for the group of COVID patients, (iii) for the group of COVID patients and the course of the disease.
- in the Material and Method section are mentioned several laboratory parameters and the chest X-Ray exam, but in the Results section are presented only a subset of the laboratory parameters and no results regarding chest X-Ray.
- The introduction could present more information about the state of the art. A more in-depth study of similar work in this field could be done.
Reviewer 3 Report
They need to review the article and correct typographical errors. In Table2, the column's header is "Enitre," it seems to be an error and should be "Entire."
Table 1 is confusing. The typography of "With the initial diagnosis of SARS-CoV-2 infection Reason" should be similar to that of "Without the initial diagnosis of SARS-CoV-2 infection Reason".
A table should be interpretable without having to read the text of the article. The meaning of abbreviations should be included in Table 1 at the bottom of the page, as done in Table 2.
In table 2, febrile neutropenia p=0.00 , please put it as p < 0.01.
In some tables, the probabilities carry 3 decimal places, and in other tables, two decimal places. Please put the p's consistently throughout the document, with identical decimals.
In table 6, I do not understand what the column "Association" means, nor how it has been calculated. You should explain it in detail.
In table 6, the units are not listed. I guess in WBC, 5.40 means 5400 WBC or 5.4 x1000WBC, but the units should be put. The units can be put on the table or at the bottom of the table. They should be put to facilitate the reading of the table.
The references should be checked, as they are inconsistent. Sometimes they are in square brackets, for example "[16]" and other times only the number appears, for example, "17".
The authors report that the most frequent signs of covid 19, such as fever, cough, dyspnoea, or gastrointestinal symptoms, few patients have covid-19. It would be interesting if the authors could calculate the predictive value of fever, dyspnea, and cough to COVID-19.
